# Accessibility and Quality of Palliative Care—Experience in Primary Health Care

**DOI:** 10.3390/medicina61010009

**Published:** 2024-12-25

**Authors:** Viljaras Reigas, Ingrida Šukienė

**Affiliations:** SMK College of Applied Science, 91199 Klaipėda, Lithuania; ingrida.puodziukaityte@googlemail.com

**Keywords:** palliative care, quality, experience, primary healthcare, accessibility

## Abstract

*Background and Objectives:* Palliative care is a very important part of medicine, aimed at ensuring an improvement in quality of life and a reduction in distressing symptoms in patients with serious, incurable, progressive diseases. The issues of the accessibility and quality of these services should be a focus for health policymakers and researchers, although it is acknowledged that a significant portion of the public has not heard about this service. For this reason, it is important to investigate the experience of the accessibility and quality of palliative care services in primary healthcare facilities. *Materials and Methods:* A quantitative study was conducted in institutions providing outpatient and inpatient palliative care services. A total of 784 patients and 219 family members participated in the study. Participants expressed their opinions through a questionnaire containing 24 statements, to which they responded by indicating their level of agreement on a Likert scale. The collected data were analyzed using statistical analysis software. *Results:* Palliative care services are widely available in large cities, but their accessibility is very limited in small towns and rural areas. Patients and their families are not familiar with the concept of palliative care, often equating it with the provision of treatment and nursing services, and they see the support of clergy as unnecessary. Although patients and their families rate the quality of the services received positively, they note shortcomings related to communication among staff. *Conclusions:* Palliative care services are provided within the primary healthcare system by specialists with qualifications regulated by legislation; however, patients do not see the need to receive assistance from clergy members. Based on the study results, it can be concluded that in Lithuania, the accessibility of palliative care is ensured in larger cities but is insufficient in smaller towns and rural areas. Patients tend to rate indicators reflecting the quality of palliative care services positively; however, they are not convinced that these services improve their quality of life.

## 1. Introduction

Palliative care services play an important role in the healthcare system, as they are intended for patients who are seriously ill and have no prospects of recovery. These patients often experience various debilitating symptoms such as nausea, vomiting, pain, breathing problems, and open wounds in various parts of the body. Such patients require continuous access to palliative care services—seven days a week, 24 h a day—regardless of the city or rural area in which the patient lives, or their financial or other circumstances. Palliative care services must be guaranteed by the country’s healthcare system, and the system must be prepared in such a way that the accessibility of these services is not limited in any aspect. Patients must receive these services immediately to ensure both their quality of life and that of their families, as well as dignified conditions for dying. Unfortunately, the limitation of access to these services is still felt in Lithuania today, despite the fact that the legal regulation of the state healthcare system ensures the provision of these services and their payment from the health insurance fund budget. The reasons for this limitation are likely the rapid aging of the population, the distribution of organizations belonging to the healthcare network, insufficient rates paid to healthcare organizations for the services provided, or a lack of awareness among healthcare professionals about the availability of palliative care services.

Global society has long been addressing issues related to aging, particularly the accessibility and quality of personal healthcare services. As the population ages, there is an inevitable increase in the number of individuals who require specialized healthcare services due to severe, incurable, progressive diseases that worsen their quality of life. The impact of these diseases is observed not only on the patients but also on their relatives and the development of the country. Although there is a wealth of research in the scientific literature focused on the analysis or prevention of chronic non-communicable diseases, there are specific areas of service availability and quality that remain insufficiently analyzed. One such area is palliative care, which is generally categorized under primary healthcare services.

The World Health Organization (2018) states in its document that palliative care encompasses a comprehensive set of elements aimed at alleviating a patient’s physical suffering and ensuring the fulfillment of social, spiritual, and psychological needs [1]. Other scientific sources indicate that palliative care is a fundamental human right and a part of integrated healthcare [2], as well as a component of primary healthcare focused on alleviating pain and symptoms troubling the patient [3]. It involves a multidisciplinary team approach aimed at improving quality of life, ensuring the continuity of services, a holistic perspective, and specialized support [4].

As can be seen from some of the definitions provided, palliative care is categorized within primary healthcare, which undoubtedly has requirements for accessibility and quality. Unfortunately, there are little data in the scientific literature regarding the availability of palliative care. Some sources criticize the focus on specific diseases or geographic locations, such as prioritizing patients with cancer or limitations imposed by residence [5]. In less developed countries, the service may not be available at all or may only be provided in a fragmented manner [6]. Additionally, the duration of service provision may be limited [7], and there is a lack of knowledge among the personnel providing these services in the field of palliative care [8].

The experience of providing palliative care varies significantly across different countries, although an analysis of the scientific literature reveals an effort to uphold the existing philosophy of palliative care worldwide. Unfortunately, it must be noted that the availability of palliative care services in various countries depends on political decisions, which are influenced by limited economic and political impact, and the progress in this area is slow [9].

In Lithuania, the culture of palliative care began to develop only at the end of 2007, when the Ministry of Health first prepared and approved the guidelines for the provision of palliative care services. By 2024, these guidelines have been revised multiple times (12 times), with the main changes focusing on both the requirements for service providers and the conditions under which patients can receive palliative care services.

To accelerate the accessibility of palliative care services and make them more available to recipients, it is crucial to identify the dimensions of service accessibility. The scientific literature identifies several dimensions: availability, acceptability, appropriateness [10], timeliness, qualified personnel [11], individual needs, health system structure and systemic factors [12], infrastructure [13], information dissemination, communication, service offerings, and the availability of specific procedures [14]. As evidenced by the data provided, the accessibility of healthcare services encompasses numerous dimensions, the entirety of which can influence the perception of service quality.

The quality of services in healthcare is perceived as a certain indicator reflecting the expectations of service recipients and is characterized by attributes such as increased competition and patient flow, as well as a high level of satisfaction with the services received [15]. According to other sources, the dimensions of quality in personal healthcare services include effectiveness, safety, accountability, continuity, timeliness, appropriateness, accessibility, and satisfaction [16,17,18].

Service quality is the part of the healthcare system focused specifically on providing healthcare services to the public. The reasons why attention is paid to the quality of individual healthcare services include the following: (1) the commitment to provide high-quality healthcare services to the public; (2) understanding what constitutes a safe, effective, and person-centered healthcare service; (3) concerns about the disparities highlighted in the standards of healthcare service delivery practices; (4) the aim to improve healthcare outcomes; (5) public expectations and the increasing demand for transparency and accountability in society; (6) the concept of universal health coverage; (7) recognition of the need to balance public and private healthcare service delivery in fragmented and mixed healthcare markets; (8) understanding that the reliability of service delivery, in preparation for potential outbreaks or other complex emergencies, is particularly important [19,20,21,22].

The quality of healthcare services emphasize W. E. Deming’s (1994) idea that a service is of high quality when its recipient enjoys it and benefits from it. They also note that the perception of the quality of healthcare services changes and depends on the position of the evaluator: different perceptions of quality and its subjective evaluation are influenced by different approaches to measuring and managing quality [20]. However, differing views on understanding quality do not prevent achieving the desired service quality.

It is important to note that ensuring the quality of individual healthcare services differs across service providers, recipients, locations, and time aspects. Such variability arises because various healthcare professionals (doctors, nurses, etc.) participate in service delivery and because of the differing needs of service recipients. Since healthcare professionals offer different methods of service delivery, the quality of services also depends on factors such as education, experience, and individual skills. Another characteristic of healthcare services is that they are provided here and now and cannot be postponed for the future. This complicates quality control, as the patient cannot evaluate the quality until the service is delivered [21].

The assumptions about the quality of healthcare organizations in the scientific literature are also linked to the absence of a common understanding of the concept of care quality. As mentioned, the definition of quality often depends on context, discipline, and perception. Evaluating and improving service quality is inseparable from the understanding that the perception of the quality of healthcare services is influenced by the specific nature of these services.

Some authors, linking the assumptions of service quality in healthcare with safety culture, recommended that organizations implement service improvement initiatives based on process redesign and safety principles, while fostering a team-oriented culture [22]. Studies of organizations conducting long-term quality and safety programs confirmed the importance of methods outlined in earlier studies and emphasized the need to improve the quality of services provided by healthcare organizations [23].

Other authors noted that many models have been developed to better understand healthcare systems and create conditions for evaluating the performance of these systems. According to the authors, many of these models regard service quality as an important goal of the healthcare system [24].

International policy documents discussing the assumptions of healthcare service quality contain information about service delivery, healthcare workers, information, medical services, funding, and leadership/management. Furthermore, in these documents, quality and safety are defined as intermediate goals of health systems, whose achievement contributes to the overall goals of the system—better healthcare quality, faster response times, and financial protection [25].

By analyzing the factors underlying the need for quality healthcare services, a systemic approach has been emphasized and pointed out that one of the most important lessons to be learned from other fields is that a common cause of errors is systemic failures [26]. Systemic, rather than human error, means that wherever people work, mistakes can happen, acknowledging the limits of their capabilities, especially when they are required to perform many tasks, particularly those that demand special focus. Florence Nightingale recognized this in 1854 when she attempted to standardize nursing practice [27]. The systemic approach contrasts with the personal view, which holds that an individual makes mistakes due to carelessness, forgetfulness, or negligence.

Other authors, who analyzed the assumptions of healthcare service quality, also emphasize the importance of applying the systemic method: (1) The systemic method acknowledges that many errors are caused by system failures, and by examining the entire system, the impact of human errors can often be reduced; (2) a systemic approach means that the quality of services is ensured by interconnected processes that influence one another and the final outcome; to ensure the quality and safety of all patient care processes, it is essential to develop a comprehensive service quality assurance program [28].

In the provision of palliative care services, the criteria for perceiving quality in the scientific literature are somewhat different. These include an integrated team, the management of pain and physical symptoms, holistic care, compassionate, empathetic, and qualified providers, timely and effective care, and the readiness of the patient and family [23,29]. Additionally, achieving established organizational and clinical quality indicators [25,30], meeting the needs of the patient and their relatives [25,31], ensuring that specialists are person-centered, facilitating communication among healthcare professionals, and having clear standards and protocols [28,32] are also emphasized.

Despite the criteria outlined for the accessibility and quality of palliative care, it can be concluded that they only partially address the existing gaps. It remains unclear what specific criteria could be used to evaluate the question, “Are palliative care services accessible and of high quality”? For this reason, it is important to investigate the experience of the accessibility and quality of palliative care services in primary healthcare facilities.

## 2. Materials and Methods

This study was conducted in healthcare institutions that provide outpatient and/or inpatient palliative care services. According to data from the third quarter of 2024, there were 124 institutions providing palliative care services in Lithuania, of which 51 organizations provided outpatient palliative care services, and 73 provided inpatient services. These organizations provide palliative care services to all residents of Lithuania, whose population, according to data from the third quarter of 2024, is 2.8 million. Primary-level institutions were chosen for two reasons: (1) according to the scientific literature sources, these services should be provided in primary-level healthcare institutions, and (2) in Lithuania, palliative care services are classified as primary-level healthcare services according to the service nomenclature.

The institutions were selected without considering their territorial distribution or the range of services offered. The only selection criterion for the institutions included in the study was that the institution must hold a license to provide inpatient and/or outpatient and/or day hospital palliative care services. Before conducting this study, all institutions providing palliative care services were contacted in writing. Seven healthcare institutions expressed their consent to participate.

In this study, healthcare institutions that agreed to participate were provided with detailed information about the planned research, its process, the research instrument, and ethical principles. The authors of the study developed a scale of 24 statements to assess patients’ experiences and/or opinions regarding the accessibility and quality of palliative care services. The research instrument was created based on the experiences and findings of other studies [1,3,4,5,6,7,8,12,18,32,33,34,35,36]. According to Setia (2017), the reliability of a questionnaire is understood as the correlation between the obtained test results and hypothetical “true” results, with one of the main characteristics being the internal consistency of the questionnaire scale, which is based on the correlation of individual questionnaire items. As mentioned, for the quantitative study—the respondents’ survey—a questionnaire consisting of 24 questions was used. The Cronbach’s alpha value for all the questions of the questionnaire is 0.932. This reliability indicator showed that the questionnaire is suitable for the further stages of research.

The research participants were asked to fill out a paper-based research instrument at their homes, which was provided to them by the principal investigator. The participants were given sufficient time to complete the research instrument. If the participants were unable to fill out the instrument at the time it was requested, another time was arranged with the principal investigator.

The following ethical principles are applied when conducting the research: (1) Conditions have been created for research that do not contradict the principle of respect for human dignity and individual rights and freedoms. Informed verbal consent was obtained from the participants to take part in the research. Participants were given the option to refuse to participate and every participant was granted the right to decide on their participation, and their refusal was respected. (2) Efforts were made to ensure that the research participants did not experience any physical, psychological, or social harm or threats to their health. Specifically, participants were given the option to refuse to answer questions that were unacceptable, unclear, or potentially distressing. (3) The research was conducted in a safe environment that ensured participants’ privacy. For example, participants in the survey were provided with a questionnaire, which they could complete in a place of their choosing, such as at home. (4) Before the research began, participants were informed in writing through the questionnaire about the key aspects of the research, the anticipated benefits, and the researcher’s contact details. In other words, all participants were provided with comprehensive information about the study, the purpose of collecting data, and the contact details of the lead researcher. (5) The information was presented in clear, understandable language, and participants’ questions were answered. Specifically, they were informed that they could ask any questions, and those questions would be answered. (6) The confidentiality of participants’ personal information and research data were ensured. The storage duration of the collected questionnaires was specified, and it was guaranteed that the participants’ identity or the identity of their organization could not be determined when publishing the research results. In other words, the information collected during the study will not be accessible to anyone other than the researchers, and the data cannot be identified by any characteristics. The research instruments used will be destroyed after the publication of the research results. Each potential study participant who was to receive palliative care services was provided with a written request for consent to participate in the study.

Participants who agreed to take part in the study completed a research instrument, which was returned to a principal investigator. A total of 784 patients and 219 relatives of patients consented to participate in this study. The main inclusion criteria for the study sample were patients receiving palliative care services and their family members providing care. There was no need to apply exclusion criteria for the study participants. The ethical aspects of the study were discussed at a meeting (18 April 2024 No. 18) of the ethics committee of SMK Higher School and met the ethical requirements for biomedical research.

The data from the quantitative study were entered into a statistical analysis program (SPSS 21 v.), which facilitated the statistical analysis—calculating frequency, mean, level of statistical significance (*p*), and standard deviation. Also, one-way ANOVA and Pearson’s r tests were used.

## 3. Results

According to publicly available information from the State Health Insurance Fund regarding palliative care service providers, it has been established that palliative care services are accessible throughout Lithuania. As of 1 October 2024, outpatient, day hospital, and inpatient palliative care services were provided by 124 service providers. Among these, some institutions provide both outpatient and inpatient palliative care services. Based on territorial principles, service providers are distributed across all counties, with providers typically located in larger cities (see Figure 1).

As seen in Figure 1, palliative care providers are spread throughout the entire territory of Lithuania; however, the concentration of these service providers is relatively low in the central-western and eastern parts of the country. It is also important to note that, according to data provided by the health insurance funds, most service providers offer only inpatient palliative care services. These data confirm that the availability of outpatient palliative care services across Lithuania is insufficient.

An analysis of the data provided by the health insurance funds revealed that 51 institutions offer outpatient palliative care services, while 73 institutions provide inpatient services. The distribution of these institutions according to the operational zones of the territorial health insurance funds is shown in Figure 2.

Based on the evaluation of the sociodemographic data of the study participants, it was determined that the statistical profile of a person receiving palliative care is as follows: a 67-year-old woman, suffering from an illness for 9 months, living with her children in a city. Meanwhile, the statistical profile of a patient’s relative can be depicted as a 43-year-old woman, married, with a higher education, and living in a city.

According to the study data, it was found that 38.5% of the study participants have cancer, 25.3% have a primary condition related to cardiovascular diseases, 16.9% have been diagnosed with a nervous system disease, and the remaining 19.3% receive palliative care services due to other identified conditions.

The results of the conducted study revealed that the concept of palliative care is not well-known among patients, and patients and their relatives tend to associate it with treatment and nursing (see Figure 3).

Analyzing the study results, it was found that in 41.2% of cases, patients were referred to receive palliative care services by their family doctor, in 22.8% cases by a doctor from an inpatient healthcare institution, and in 36.0% of cases, patients sought help on their own initiative or were recommended by acquaintances. It was also determined that the average waiting time for palliative care, measured in days from the time of referral, was 2.8 days.

An analysis of the collected data revealed that a doctor visits a patient in need of palliative care on average once a week, while a nurse, nursing assistant, and physiotherapist visit 2–3 times. A social worker visits once every two weeks, and patients indicated that they did not wish to receive services from a medical psychologist or a clergy member.

The analysis of the research results revealed that 74.5% of patients are satisfied with their first meeting with the palliative care team members. When evaluating the reasons for satisfaction, it was found that 42.3% appreciated the attention shown to them, 54.3% felt they were valued in society, and 16.8% indicated that communication with the palliative care team members increased their trust in healthcare. On the other hand, when evaluating the negative experiences of patients and their relatives, the study participants tended to emphasize verbal body language (28.7%), indifference to the patients’ and their relatives’ problems (19.4%), and miscommunication (33.2%).

During the study, patients were asked to evaluate factors that could affect the quality of the palliative care services they received. As seen from the results obtained, patients rated the alignment of services with their expectations the lowest (mean = 3.6; SD = 0.8) as well as communication among team members (mean = 3,8; SD = 0.7) (see Figure 4).

During the study, the effectiveness of palliative care services was analyzed, and patients or their relatives were asked to indicate how often in the past week the patient experienced distressing physical symptoms. The results revealed that 73.2% of patients experienced pain in the past week, 10.2% experienced nausea, 91.6% felt weakness, 8.5% had to visit the emergency department due to a worsening condition, and 39.8% indicated that the symptomatic treatment they received was effective. As can be seen from the data obtained, in many cases, one of the main goals of palliative care—reducing physical symptoms that negatively affect the patient—remains unachieved.

Data from various scientific sources indicate that one of the primary goals of palliative care is to improve the quality of life for patients and their relatives [3,4,5,6,7]. As the results of the conducted study show, palliative care services in Lithuania do not fully achieve this goal—the study participants rated the area “services improve quality of life” the lowest (mean score—3.6). Additionally, the area of service timeliness was rated relatively low (mean score—3.9), suggesting that participants are more inclined to disagree than agree that they received palliative care services in a timely manner. The average scores for other areas reflecting the quality of palliative care services exceeded 4, which suggests that participants evaluate these areas positively (see Figure 5).

This study analyzed the ability and empowerment of patients’ relatives to participate in the palliative care process. As seen from the data, a significant portion of patients’ relatives avoid participating in the provision of palliative care services because they feel that they lack the necessary knowledge (56.7%) or skills (41.2%), or fear they might harm their loved one (65.8%). Participants also noted that palliative care professionals often do not encourage involvement in the service delivery process, or such encouragement is minimal. It was found that only 38.5% of relatives have been trained in how to act in specific situations with their loved ones. These findings reveal that in Lithuania, there is an insufficient culture promoting the inclusion of relatives in the palliative care process.

During the study, when patients were asked to rate the palliative care services received on a scale from 0 to 10, where 0 means very poor and 10 means excellent, it was found that patients tended to rate the services positively, with an average score of 8.8. The study participants were asked to justify their choices, which were most often related to the ability to alleviate negative physical symptoms and communication.

## 4. Discussion

As seen from the results of this study, palliative care services in Lithuania are a relatively new area of healthcare compared to other European and world countries, where the culture of these services dates back to the 20th century. Despite this, Lithuania is expanding the accessibility of palliative care services and increasing the funding allocated from the health insurance fund budget. However, despite state planning and available opportunities, there is still a shortage of service providers, especially in outpatient palliative care services. The results of this study reveal that the network of service providers is insufficient, leading to a lack of these services in smaller towns and rural areas. For these reasons, we can assume that society is inadequately informed about the concept of palliative care, and these services are often confused with nursing care services.

There is particularly high attention given to the quality of healthcare services in society, but there is a lack of elements that reflect quality for specific healthcare services. Palliative care is no exception. As seen from the results of this study, indicators reflecting quality could include the ability of service providers to alleviate the symptoms that burden patients, but attention must also be paid to indicators such as the ability to implement the palliative care plan or the level of its implementation. Unfortunately, this practice does not yet exist in Lithuania.

The results of this study partially align with findings from earlier studies and add new data. In Lithuania, palliative care services, as in some other countries, are provided in both general and specialized hospitals and are available in three forms: outpatient (provided in patients’ homes), day hospital, and inpatient. However, when evaluating the geographical distribution of service providers, issues of accessibility in smaller towns or rural areas can be observed. The fact that palliative care can be provided in outpatient and/or inpatient settings, offering patients the freedom of choice, is also highlighted by studies conducted in Germany [33], the USA [34], and Australia [37].

As seen from the data obtained during the study, three out of ten patients receiving palliative care services have been diagnosed with cancer. These results partially contradict earlier studies [35,38,39,40], which indicate that palliative care is mostly provided for cancer patients.

The analysis of the study data shows that patients, although receiving palliative care aimed at reducing physical suffering, still experience the effects of distressing symptoms and, in some cases, are forced to visit the emergency department. These data nonetheless demonstrate that palliative care does not achieve its goal of reducing physical suffering in all cases [1,3,4].

Until now, patients and their relatives have not been thoroughly familiarized with the concept of palliative care, as they tend to focus less on meeting social and spiritual needs, prioritizing nursing and treatment actions instead. These results are aligned with findings from a study conducted in the USA, which stated that 71.6% of participating adults had never heard of palliative care [36].

Although the palliative care services provided are rated positively by patients and their relatives, they do not fully meet their expectations. The results of a study conducted in 2024 partially complement the findings of this study and indicate that the primary needs of patients are related to physical, emotional, and social well-being, as well as access to information and independent decision-making. The authors emphasize the importance of ensuring that the services provided align with patients’ expectations and address the challenges associated with increased caregiver burden and maintaining a healthy work–life balance [41].

One study conducted in Brazil [19] emphasized the communication among team members providing palliative care and its impact on the quality of those services. The results of the study indicate that inadequate communication affects the quality of the services provided. An analysis of the results revealed that patients often notice the lack of effective communication among specialists delivering palliative care, which in turn may be linked to patients and their relatives not being fully convinced of the services’ effectiveness in improving their quality of life.

Considering the results of this study, it would be advisable to expand the scope of the research to assess the experience of palliative care service delivery from the perspective of service providers. It can be assumed that such studies could help provide a more comprehensive understanding of the planning, monitoring, and evaluation measures for palliative care quality. It would also be beneficial to include the opinions of palliative care professionals in such studies, particularly regarding the factors that shape quality.

Despite the future possibilities for research, the results obtained help to understand the novelty of palliative care services in Lithuania, as well as to compare the factors influencing quality with the results of studies conducted worldwide and in Europe. As seen from the data and their comparison, the provision of palliative care in Lithuania faces the same challenges encountered in other countries.

A limitation of this study is that a relatively small number of patients receiving palliative care, and their family members were surveyed. Future studies should aim to survey a larger number of participants, with a broader geographical distribution (across the entire country of Lithuania).

## 5. Conclusions

Palliative care services are provided within the primary healthcare system by specialists with qualifications regulated by legislation; however, patients do not see the need to receive assistance from clergy members. Based on the study results, it can be concluded that in Lithuania, the accessibility of palliative care is ensured in larger cities but is insufficient in smaller towns and rural areas. Patients tend to rate indicators reflecting the quality of palliative care services positively; however, they are not convinced that these services improve their quality of life.

## Figures and Tables

**Figure 1 medicina-61-00009-f001:**
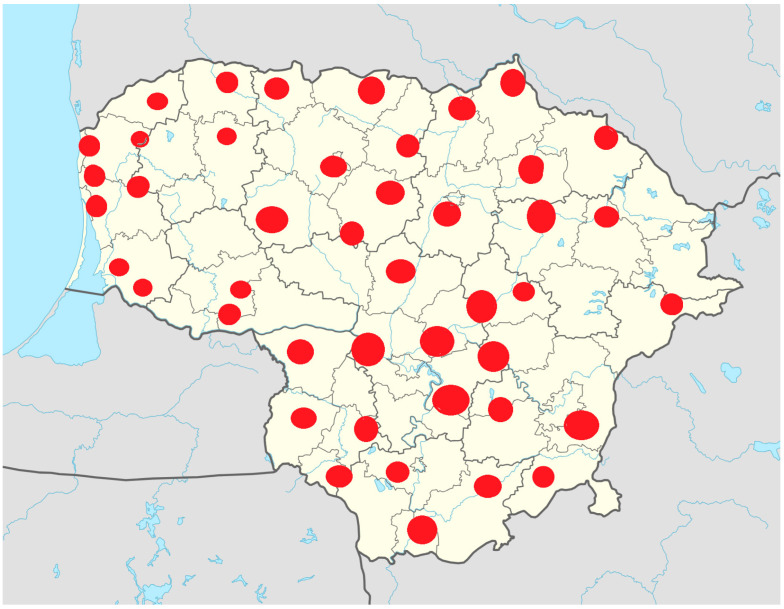
Concentration of institutions providing palliative care services in the territory of the Republic of Lithuania. Source: prepared by the author based on data from the State Health Insurance Fund (2024).

**Figure 2 medicina-61-00009-f002:**
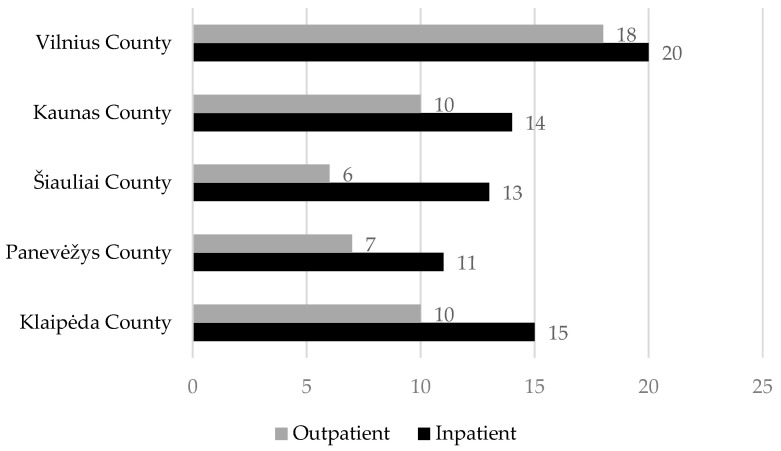
Distribution of institutions providing palliative care services according to the operational zones of the territorial health insurance funds (the specified number of organizations). Source: compiled by the author based on data from the State Health Insurance Fund (2024).

**Figure 3 medicina-61-00009-f003:**
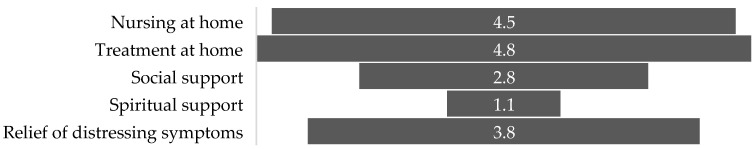
The understanding of palliative care is expressed by patients and their relatives (the mean rating is indicated).

**Figure 4 medicina-61-00009-f004:**
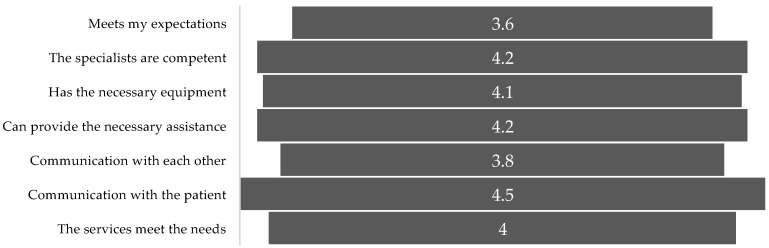
Patients’ opinions on factors influencing their perception of the quality of palliative care. (the mean rating is indicated).

**Figure 5 medicina-61-00009-f005:**
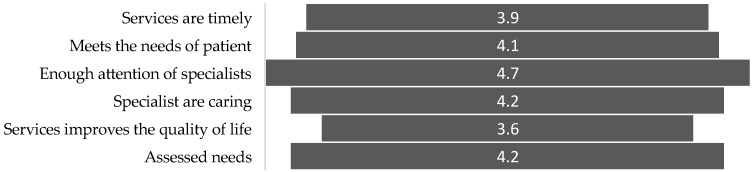
Quality elements of palliative care (the mean rating is indicated).

## Data Availability

The original contributions presented in the study are included in the article, further inquiries can be directed to the corresponding author/s.

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
