# Peer review of "Accessibility and Quality of Palliative Care—Experience in Primary Health Care"

_medicina, 2024, doi:10.3390/medicina61010009_

Round 1
Reviewer 1 Report
Comments and Suggestions for Authors
Great work by the authors.
The suggestion I have is:
1. For figures 2-4: would suggest explaining what those numbers in the bars mean? For example: nursing at home 4.5 - what is the exact meaning of 4.5? Does it mean that the mean rating was 4.5? this needs to be explained clearly.
2. "As indicated by the results, palliative care services do not achieve their primary goal of improving the quality of life for patients and their relatives"- can you explain how this conclusion is made, I understand the score for this is 3.6 (on the bar)- so is the cut-off < 4 to call that particular thing was not achieved/significant. This needs to be clarified.
Author Response
Dear Reviewer,
Thank you for your comments and review of our research results. We fully agree with your remarks and have made the necessary changes.
Comment No 1: "1. For figures 2-4: would suggest explaining what those numbers in the bars mean? For example: nursing at home 4.5 - what is the exact meaning of 4.5? Does it mean that the mean rating was 4.5? this needs to be explained clearly. "
Answer for the Comment No 1: We have updated the figure numbering and indicated for each figure that the mean ratings of the responses are presented.
Comment No 2: "As indicated by the results, palliative care services do not achieve their primary goal of improving the quality of life for patients and their relatives"- can you explain how this conclusion is made, I understand the score for this is 3.6 (on the bar)- so is the cut-off < 4 to call that particular thing was not achieved/significant. This needs to be clarified.
Answer for the Comment No 2: The research paragraph in which this statement is presented is clarified: "Data from various scientific sources indicates that one of the primary goals of palliative care is to improve the quality of life for patients and their relatives [3-7]. As the results of the conducted study show, palliative care services in Lithuania do not fully achieve this goal – the study participants rated the area "Services improve quality of life" the lowest (mean score - 3.6). Additionally, the area of service timeliness was rated relatively low (mean score - 3.9), suggesting that participants are more inclined to disagree than agree that they receive palliative care services in a timely manner. The average scores for other areas reflecting the quality of palliative care services exceeded 4, which suggests that participants evaluate these areas positively (see Figure 5)."
All changes in the abstract are highlighted in yellow.
Thank you once again for your comments and suggestions.
We wish you a good day!
Authors
Reviewer 2 Report
Comments and Suggestions for Authors
Dear Authors:
I have reviewed your paper entitled "Accessibility and Quality of Palliative Care – Experience in Primary Health Care." This quantitative study aims to analyze the practice of providing palliative care in primary healthcare and to identify the accessibility and quality indicators used in practice. It was developed with 784 patients and 219 family members participated in the study of institutions in Lithuania that provided outpatient and inpatient palliative care services. The data instrument used was a questionnaire that contained 24 statements.
In conclusion, the authors refer to an existent disparity in accessibility of palliative care when comparing the larger cities and smaller towns and rural areas.
First, I would like to commend you on your work and your choice of this significant topic. The paper addresses an important area in palliative care; however. As an external reviewer, I have identified some aspects that could strengthen the paper. I would like to offer the following suggestions and reflections for improvement:
1) Abstract: Please standardize the aims in the abstract (page1 line 12-13 and text (page 4 line 155-156);
2) Keywords: please considered insert more keywords as: Primary Healthcare; Accessibility.
3) Introduction: Please insert some citation in the first paragraph (is missing some reference to justify the ideas that you used).
4) Material and Methods: Could you give us a picture of your reality (population rates; number of palliative care institutions) and in results please insert only results of the study.
5) You mentioned the possibility to participants withdraw at any stage, who you assure that? Did you use codification data?
6) About data instrument: did you use paper format? and tell us more about the data collecting process (Who dis; how and where)? The principal researcher or healthcare professionals in the field?
7) Do you have a code from ethic committee approval?
8) Please tell us what statistic program you used. And refer the statistic tests you selected and used for your study.
9) Did you have exclusion criteria for the sample?
10) Figure 2 (page 6); figure 3, 4 (page 7); is blurred.
11) Discussion: Please insert a paragraph with study limitation.
I have no additional suggestions currently, and I wish you the best with the publication of your paper!
Best regards.
Author Response
Dear Reviewer,
Thank you for your comments and suggestions. We sincerely appreciate them and have responded to all of them. Below, we provide our responses, references, and all the changes made.
All the changes made are highlighted in yellow.
Comment No 1: Abstract: Please standardize the aims in the abstract (page1 line 12-13 and text (page 4 line 155-156).
Answer to the Comment No 1: We fully standartized the aims in the abstract and text. (lines 12-14 and 177-178).
Comment No 2: Keywords: please considered insert more keywords as: Primary Healthcare; Accessibility.
Answer to the Comment No 2: We updated keywords. (line 30).
Comment No 3: Introduction: Please insert some citation in the first paragraph (is missing some reference to justify the ideas that you used).
Answer to the Comment No 3: We updated with the citations in first paragraph. (lines 33-49).
Comment No 4: Material and Methods: Could you give us a picture of your reality (population rates; number of palliative care institutions) and in results please insert only results of the study.
Answer to the Comment No 4: We updated the part of Material and Methodology with the picture of our reality (population rate, number of palliatyive care institution). In results we left only the results of the study. (lines 181-185).
Comment No 5: You mentioned the possibility to participants withdraw at any stage, who you assure that? Did you use codification data?
Answer to the Comment No 5: We clarified the expression as follows: ". Participants were given the option to refuse to participate and every participant was granted the right to decide on their participation, and their refusal was respected". (lines 217-219).
Comment No. 6: About data instrument: did you use paper format? and tell us more about the data collecting process (Who dis; how and where)? The principal researcher or healthcare professionals in the field?
Answer to the Comment No 6: we updated withe the paragraph about how the date were collected. (line 209-213).
Comment No 7: Do you have a code from ethic committee approval?
Answer to the Comment No 7: The ethical aspects of the study were discussed at a meeting (18/04/2024 No. 18) of the ethics committee of SMK Higher School and met the ethical requirements for biomedical research. This is inserted to the Material and Methods part. (lines 243-246)
Comment No 8: Please tell us what statistic program you used. And refer the statistic tests you selected and used for your study.
Answer to the Comment No 8: Information about statistical programm is updated to the Material and Method part (SPSS, v. 21). Also we update information about statistic tests. (lines 249-252).
Comment No 9: Did you have exclusion criteria for the sample?
Answer to the Comment No 9. We didint use exlusion criteria for the sample. We updated part of Material and Method about this (line 245).
Comment No 10: Figure 2 (page 6); figure 3, 4 (page 7); is blurred.
Answer to the Comment No 10: We improved the Figures.
Comment No 11: Discussion: Please insert a paragraph with study limitation.
Answer to the Comment No 11: We updated the part of Discussion about the limitation of this study (lines 425-428).
Thank you and have a good day!
Best wishes,
Authors
Round 2
Reviewer 2 Report
Comments and Suggestions for Authors
Dear Authors,
Thank you for your revisions and response.
I believe the content is now clearer for readers. While the figures have not improved significantly, I think the editorial board may provide further assistance.
Wishing you the best for your paper's publication and happy holidays.
Best regards,
Author Response
Dear Reviewer,
thank you for your evaluation.
All the best,
Authors